# Turbulence Measurements with Dual-Doppler Scanning Lidars

**Alfredo Peña *** and **Jakob Mann**

DTU Wind Energy, Technical University of Denmark, 4000 Roskilde, Denmark; jmsq@dtu.dk
* Correspondence: aldi@dtu.dk

**Abstract:** Velocity-component variances can be directly computed from lidar measurements using information of the second-order statistics within the lidar probe volume. Specifically, by using the Doppler radial velocity spectrum, one can estimate the unfiltered radial velocity variance. This information is not always available in current lidar campaigns. The velocity-component variances can also be indirectly computed from the reconstructed velocities but they are biased compared to those computed from, e.g., sonic anemometers. Here we show, for the first time, how to estimate such biases for a multi-lidar system and we demonstrate, also for the first time, their dependence on the turbulence characteristics and the lidar beam scanning geometry relative to the wind direction. For a dual-Doppler lidar system, we also show that the indirect method has an advantage compared to the direct one for commonly-used scanning configurations due to the singularity of the system. We demonstrate that our estimates of the radial velocity and velocity-component biases are accurate by analysis of measurements performed over a flat site using a dual-Doppler lidar system, where both lidars stared over a volume close to a sonic anemometer at a height of 100 m. We also show that mapping these biases over a spatial domain helps to plan meteorological campaigns, where multi-lidar systems can potentially be used. Particularly, such maps help the multi-point mapping of wind resources and conditions, which improve the tools needed for wind turbine siting.

**Keywords:** atmospheric flow; scanning lidars; spectra; turbulence; variances

## 1. Introduction

Scanning Doppler wind lidars (hereafter scanning lidars) are lidars that, depending on the particular need, can probe the atmosphere at almost any given scanning configuration. They have been employed within a wide range of applications including weather and climate monitoring [1,2], atmospheric boundary-layer and turbulence studies [3–5], and wind energy [6,7]. Within the latter application, scanning lidars have become intensively used for different purposes in recent years. It is now common to employ such lidars in wind resource assessment [8,9], power curve performance measurements [10], site conditions [11,12], wake analysis [13,14], and wind forecasting [15].

A dual-Doppler scanning lidar system consists of two scanning lidars separated from some tens of meters and even up to kilometers [8]. Such a system is chosen over other configurations because of different reasons. First, one can use a single scanning lidar to estimate both wind speed and wind direction but we need to measure in more than one direction and assume that the wind is homogeneous within the scanned volume, which is an assumption only valid for flat and homogeneous terrain. Thus, the uncertainty (and sometimes the accuracy) of estimating the horizontal velocity components with such a configuration is too high, particularly when measuring at the kilometer range. Second, in wind energy we are mostly interested in the horizontal velocity components (wind speed and direction), the longitudinal turbulence, i.e., the longitudinal velocity-component variance, and the vertical wind shear, which are parameters that can be retrieved accurately with a dual-system configuration (here

we focus our attention on the accuracy of turbulence estimates with such systems). Third, scanning lidars are expensive and, thus, acquisition and maintenance of a triple system, which in principle can be used to estimate the three velocity components and their variances, are not insignificant issues. Fourth, measuring with multi-lidar systems is not trivial and logistically a difficult matter, particularly for setups where their beams are synchronized both in space and time.

Dual-Doppler scanning lidar systems were recently used in the main experimental campaigns within the New European Wind Atlas project [9,16]. In the RUNE campaign [8], the dual setup provided detailed information with regards to the effect of the land and the sea-to-land change of roughness on the wind. In the Perdigão campaign [17,18], multi-lidar systems were used to study the wake after a solitary wind turbine on a ridge [19–21], and trapped lee-waves after two ridges under stable conditions and the observations were compared to results from numerical modelling [22,23]. Menke et al. [24] used three pairs of scanning lidars in the same experiment to characterize how recirculation after the ridges depends on atmospheric stability and position in the landscape. Scanning lidars were mounted on platforms on two masts separated by 4.25 km to investigate the dynamics of flow in a horizontal plane over a flat, but heterogeneous landscape in the Østerild Balconies experiment [25].

It is important to note that in most of the previous studies, the works concentrated in the analysis of the flow either by looking at the radial velocity (also known as the line-of-sight velocity), which is the basic output of a lidar scan, or at the velocity components, which were reconstructed from radial velocities using different methodologies. We believe that this is partly because the understanding of atmospheric turbulence and how lidars probe the atmosphere are complex matters. We show, for the first time, how to compute the bias of velocity-component variances using radial velocity measurements from a multi-lidar system, in relation to those variances from an ideal anemometer. We also show for the first time the dependence of these biases on turbulence characteristics, scanning geometry, and lidar characteristics.

Here, we start with a general background on lidars (Section 2), where we first demonstrate their capabilities to measure atmospheric turbulence (Section 2.1). Then we illustrate two ways of estimating turbulence with a dual-Doppler scanning system (Section 2.2), where their advantages and disadvantages are discussed. Then we describe an experimental campaign in which a dual system measured during slightly more than one month over a volume close to a sonic anemometer at 100 m (Section 3); this campaign is the basis of our analysis. Results with regards to observed biases of radial velocity variances and biases of velocity-component variances between measurements from the sonic anemometer and the dual system are shown in Sections 4.3 and 4.4 together with theoretical estimations of these biases. Maps have been constructed showing the spatial variability of the bias of the velocity-component variance by a dual system in Section 4.6. Finally, Discussion and Conclusions are drawn in the last two sections.

## 2. Background

### 2.1. Generalities

Turbulence information from lidar(s) measurements can be retrieved using two methods. For the first (direct) method, we compute second-order statistics of the three velocity components based on the second-order statistics of the lidar(s) radial velocities. This has the potential advantage that the computed second-order statistics, e.g., the velocity-component variances, can be unbiased, when compared to the variances from an ideal instrument, if the second-order statistics of the lidar(s) radial velocities account for the averaging (filtering) effect of the measurement volume. For the second (indirect) method, we compute second-order statistics of the three velocity components directly from the reconstructed velocity components, which have been already computed based on the lidar(s) radial velocities. The latter method has the disadvantage that the computed second-order statistics are, if not all most of the times, biased as we will explain later. However, as we will also show, the indirect

method also has advantages compared to the direct method, particularly when turbulence is estimated using a dual-scanning system.

In order to understand the origin of these biases and filtering effects, we can start by assuming that the measurements are performed with a single scanning lidar (see Figure 1), which is measuring the wind $\mathbf{v} = (u, v, w)$, where $u$, $v$, and $w$ are the velocity components along the horizontal axes $x$- and $y$-, and the vertical axis $z$, respectively, at a distance $d_f$ with a beam in the direction of the unit vector $\mathbf{n}$,

$$\mathbf{n}(\phi, \theta) = (\cos\theta\cos\phi, \sin\theta\cos\phi, \sin\phi), \tag{1}$$

where $\theta$ is the azimuth angle in the $x$-$y$ plane and $\phi$ the elevation (tilt) angle with respect to the same plane. For simplicity, we first assume that the wind is aligned with the $x$-axis. The radial velocity of a scanning lidar can be expressed as [26]:

$$v_r(\phi, \theta, d_f) = \mathbf{n}(\phi, \theta) \cdot \mathbf{v}\left[\mathbf{n}(\phi, \theta)d_f\right]. \tag{2}$$

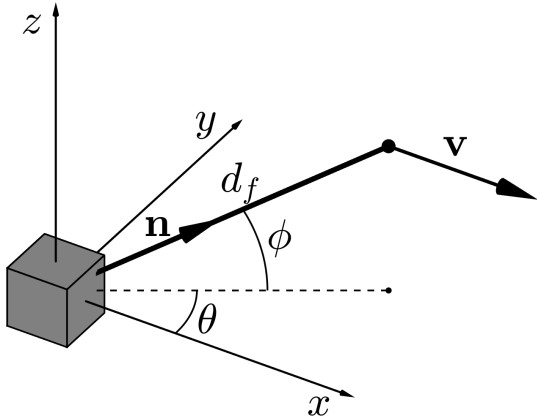

**Figure 1.** Sketch of a scanning lidar measuring the wind $\mathbf{v}$ at a distance $d_f$ with a beam in the direction of the unit vector $\mathbf{n} = \mathbf{n}(\theta, \phi)$.

Equation (2) ignores any averaging along the beam. However, we might consider all velocities within the probe volume and, thus, a weighted average radial velocity along the lidar beam will be [26]:

$$\tilde{v}_r(\phi, \theta, d_f) = \int_{-\infty}^{\infty} \varphi(s)\mathbf{n}(\phi, \theta) \cdot \mathbf{v}\left[\mathbf{n}(\phi, \theta)(d_f + s)\right] ds, \tag{3}$$

where $\varphi$ is the weighting function that depends on the lidar type and characteristics, and $s$ the distance along the beam from the point of interest. Such averaging affects the spectrum of the lidar radial velocity (see e.g., [11]):

$$F_{v_r}(k_1) = n_i n_j \iint |\hat{\varphi}(\mathbf{k} \cdot \mathbf{n})|^2 \, \Phi_{ij}(\mathbf{k}) dk_2 dk_3, \tag{4}$$

where $\hat{\varphi}$ is the Fourier transform of $\varphi$, $\Phi_{ij}$ the spectral velocity tensor, i.e., the three-dimensional spatial statistics of velocity fluctuations, and $\mathbf{k} = (k_1, k_2, k_3)$ the wave vector. Equation (4) shows that the lidar radial velocity spectrum is rather similar to the one-point spectra of the velocity components,

$$F_{ij}(k_1) = \iint \Phi_{ij}(\mathbf{k}) dk_2 dk_3, \tag{5}$$

with the addition of the effect of the probe volume and the beam direction. We will be using the model of Mann [27] (hereafter Mann model) to describe $\Phi_{ij}$. The Mann model contains three parameters (know as Mann parameters), besides $\mathbf{k}$, related to the dissipation rate $\alpha\epsilon^{2/3}$, the length scale of turbulence $L$, and anisotropy $\Gamma$.

Let us first assume that the lidar measures over a volume so small that we can neglect any averaging, i.e., $\hat{\varphi}(\mathbf{k} \cdot \mathbf{n}) = 1$. Figure 2a shows the difference between the $u$- and $w$-velocity spectra, and the radial velocity spectrum from a lidar whose beam is tilted from the $x$-$y$ plane. For zero tilt, $F_{v_r} = F_u$ and at a 90° tilt, $F_{v_r} = F_w$.

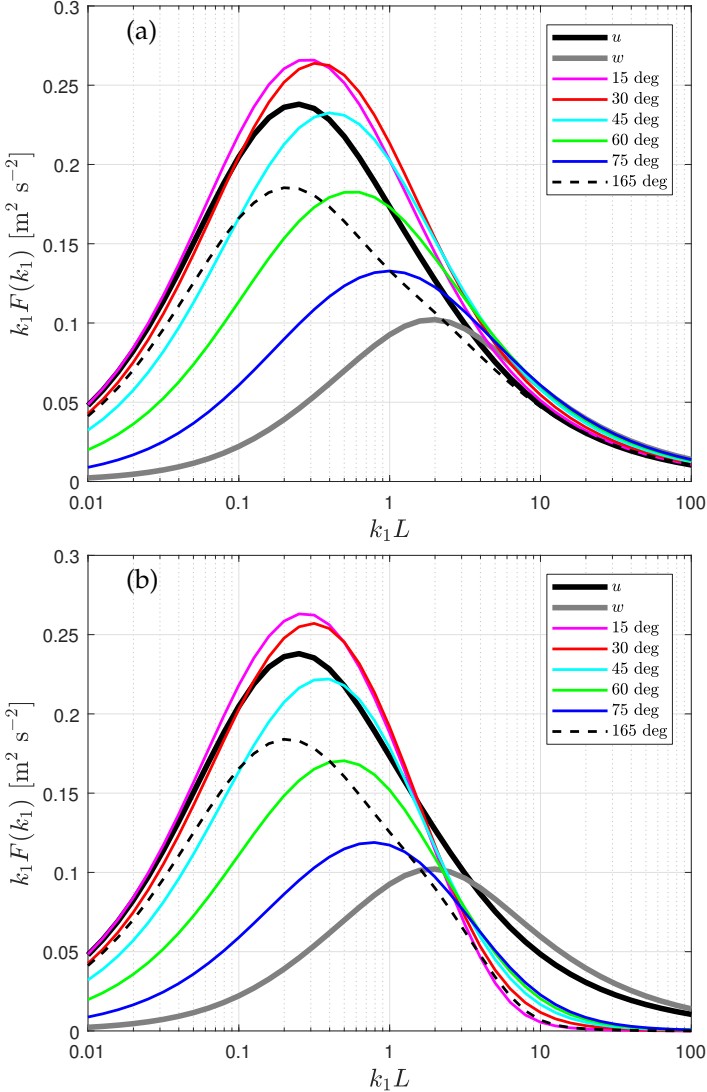

**Figure 2.** Simulated $u$- and $w$-velocity spectra and radial velocity spectra of a lidar with (**a**) and without (**b**) the effect of filtering for different elevation angles with respect to the mean wind for $\alpha \epsilon^{2/3} = 0.1$ m$^{4/3}$ s$^{-2}$, $L = 50$ m, and $\Gamma = 3$. For the case of filtering (b), $z_R/L = 0.5$.

We observe that for some relatively low tilt angles, the radial velocity spectrum contains more energy than the $u$-velocity spectrum. Following Equation (4), we notice that $F_{v_r}$ has contributions (also sometimes referred to as contamination) from $\Phi_{11}, \Phi_{22}, \Phi_{33}, \Phi_{12}, \Phi_{23}, \Phi_{13}$, i.e., of the $u$-, $v$- and $w$-velocity spectra and $uv$-, $vw$-, and $uw$- cospectra. The Mann model (uniform shear version) assumes $F_{12} = F_{23} = 0$, so the spectra of the radial velocity of a tilted lidar beam aligned with the wind is at least, in the light of the Mann model, a mix of $u$-, and $w$-spectra and $uw$-cospectra. Note that at 15° tilt, $\Phi_{13}$ 'positively' contributes to $F_{v_r}$ and the opposite behavior occurs at 165°.

Figure 2b shows the same velocity spectra as Figure 2a but for a lidar with a weighting function of the type

$$\hat{\varphi}(k_1) = \text{sinc}^2(k z_R/2), \qquad (6)$$

where $z_R$ characterizes the length of the probe volume. This is a weighting function typical of pulsed lidars (see e.g., [28]), where $z_R$ is half the length of a rectangular pulse. It is clearly shown the effect of filtering, particularly for the high wave numbers.

For the filtering case, i.e., Figure 2b, it is seen that, although filtered, the radial velocity spectrum for the first two elevation angles still peaks higher than the $u$-velocity spectra; there is thus a given $z_R/L$ value for which the radial velocity variance $\sigma_{v_r}^2$ is equal to the $u$-velocity variance $\sigma_u^2$, although $z_R/L \neq 0$. There is also a range of $z_R/L$ values where $\sigma_{v_r}^2/\sigma_u^2 > 1$ for relatively low tilt angles.

*2.2. Dual-Doppler Scanning System*

Here we present two ways to estimate horizontal velocity variances with a dual-Doppler system. When evaluating observations and estimations of variances from an experimental campaign (see Section 4), only the second method will be used due to the level of information of the lidar measurements. However, the first method is introduced here to provide the reader with a wider understanding of lidar turbulence measurements.

2.2.1. Direct Velocity and Variance Computations

From Equations (1) and (2) and assuming a zero vertical velocity component, we can estimate the two horizontal velocity components with a dual-Doppler scanning system as,

$$\underbrace{\begin{bmatrix} v_{r_1} \\ v_{r_2} \end{bmatrix}}_{\mathbf{vr}} = \underbrace{\begin{bmatrix} \cos\theta_1\cos\phi_1 & \sin\theta_1\cos\phi_1 \\ \cos\theta_2\cos\phi_2 & \sin\theta_2\cos\phi_2 \end{bmatrix}}_{\mathbf{M}} \underbrace{\begin{bmatrix} u \\ v \end{bmatrix}}_{\mathbf{v}}, \tag{7}$$

where the subindexes 1 and 2 refer to the two scanning lidars, and its variances as:

$$\underbrace{\begin{bmatrix} \sigma_{v_{r_1}}^2 \\ \sigma_{v_{r_2}}^2 \end{bmatrix}}_{\mathbf{S}} = \underbrace{\begin{bmatrix} \cos^2\theta_1\cos^2\phi_1 & \sin^2\theta_1\cos^2\phi_1 \\ \cos^2\theta_2\cos^2\phi_2 & \sin^2\theta_2\cos^2\phi_2 \end{bmatrix}}_{\mathbf{P}} \underbrace{\begin{bmatrix} \sigma_u^2 \\ \sigma_v^2 \end{bmatrix}}_{\mathbf{Q}}, \tag{8}$$

where we assume that the $uv$-covariance is zero. The velocity components can then be computed by

$$\mathbf{v} = \mathbf{M}^{-1}\mathbf{vr} \tag{9}$$

and the velocity variances by

$$\mathbf{Q} = \mathbf{P}^{-1}\mathbf{S}. \tag{10}$$

The velocity variances will be unbiased if the measured radial velocity variances are unfiltered. This can be achieved by avoiding the effects of averaging within the measurement volume using information of the Doppler radial velocity spectrum [26,28].

The systems in Equations (7) and (8) show that the estimations of $\mathbf{v}$ and $\mathbf{Q}$ depend on how invertible the geometrical matrices $\mathbf{M}$ and $\mathbf{P}$ are. For the simple case where $\phi_1 = \phi_2 = 0°$, one can notice that $\det(\mathbf{M})$ is highest at $\theta_1 = \theta_2 = 45°$ and that $\mathbf{M}$ is singular for small or equal azimuths with respect to the axis between both lidars, e.g., $\theta_1 = 0°$ and $\theta_2 = 180°$. For $\mathbf{P}$ results are similar except that the matrix also becomes singular for supplementary angles. This is because of the cosine and sine of the azimuths are squared, so the system cannot distinguish the contributions of the $u$- and $v$-components. Figure 3 shows within a spatial horizontal domain both $\det(\mathbf{M})$ and $\det(\mathbf{P})$ for a dual-Doppler scanning system. The direct methods for estimating velocity variances will thus show high random errors in the areas where we can estimate the velocity components with low random errors.

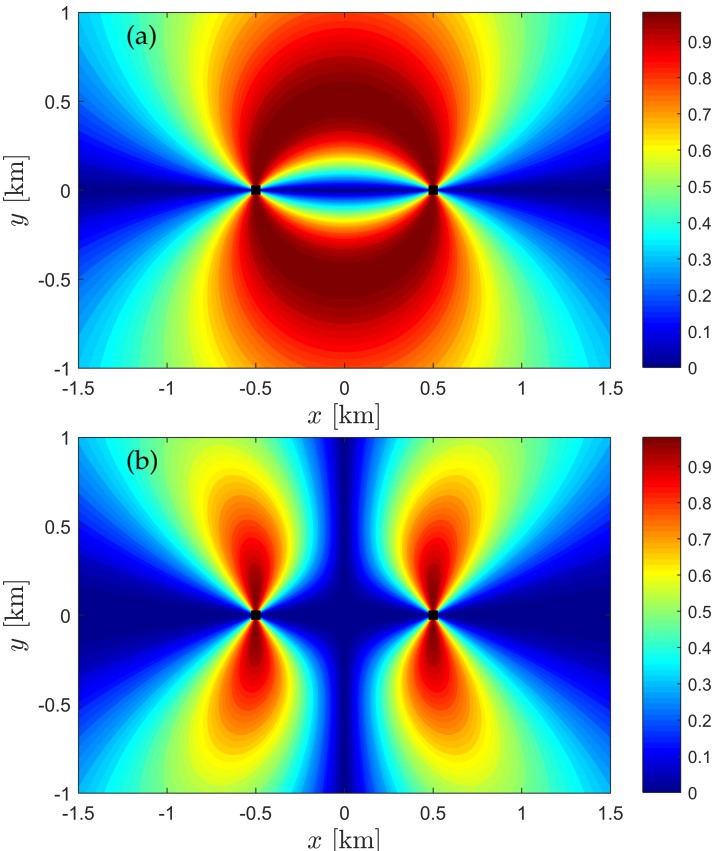

**Figure 3.** Spatial distribution of the determinant of **M** (**a**) and **P** (**b**) for a dual-Doppler scanning system with $\phi_1 = \phi_2 = 0°$ assuming the wind is aligned with the *x*-axis.

### 2.2.2. Indirect Variance Computations

As mentioned in Section 2.1, computing the second-order statistics from reconstructed velocity components of a scanning lidar leads, in most of the cases, to biases due to filtering and contamination effects. The relevant question is whether or not we can estimate these biases accurately.

The Reynolds stresses in the Cartesian coordinate system can then be computed as

$$\langle u_i' u_j' \rangle = N_{i\alpha} \langle v_{r,\alpha}' v_{r,\beta}' \rangle N_{j\beta}, \tag{11}$$

where

$$\langle v_{r,\alpha}' v_{r,\beta}' \rangle = n_i^\alpha n_j^\beta \int \hat{\varphi}(\mathbf{k} \cdot \mathbf{n}^\alpha) \hat{\varphi}(\mathbf{k} \cdot \mathbf{n}^\beta) \Phi_{ij}(\mathbf{k}) d\mathbf{k}, \qquad \text{(no summation over } \alpha \text{ and } \beta) \tag{12}$$

is the covariance matrix of radial velocities, being $\alpha$ and $\beta$ subscripts indicating the lidar numbering, and assuming a dual-lidar system

$$\mathbf{N} = \left[ \begin{array}{ccc} \cos\theta_1 \cos\phi_1 & \sin\theta_1 \cos\phi_1 & \sin\phi_1 \\ \cos\theta_2 \cos\phi_2 & \sin\theta_2 \cos\phi_2 & \sin\phi_2 \end{array} \right]^+, \tag{13}$$

where "+" is the pseudoinverse of the matrix. Note that by using two lidars only, Equation (11) gives the Reynolds stresses assuming that the velocity component perpendicular to the plane spanned by the two beams is zero.

Equation (11) shows that the Reynolds stresses are function of the covariance matrix between radial velocities from the number of lidars available and the scanning geometry, and Equation (12) shows that this covariance matrix is a function of the turbulence characteristics, the probe volume of the

lidars, and the scanning geometry. The idea is that in the case where no averaging effects are significant, i.e., $\hat{\varphi}(\mathbf{k} \cdot \mathbf{n}) \approx 1$, the dependence on the scanning geometry of the covariance matrix is compensated by the scanning geometry dependence in Equation (11) so that velocity-component variances computed with the latter expression are unbiased and a function of the turbulence characteristics only. This is the case of a sonic anemometer with three pairs of transducers, which can be seen as a multi-lidar scanning system with three beams.

For the particular case of a dual-Doppler scanning system, computing the Reynolds stresses is only slightly easier compared to a multi-lidar system; the covariance matrix of radial velocities is of size $2 \times 2$. The complexity of the calculation involves the computation of each term in the covariance matrix (Equation (12)).

Further, for a dual-Doppler system scanning at a point, i.e., a system whose probe volume is so small that no averaging occurs, unbiased velocity-component variances cannot be obtained using Equation (11). This is because we need to have at least a $3 \times 3$ covariance matrix to solve the system in Equation (12) without the need of assumptions regarding relations between covariances. But we know that these relations are dependent on the turbulence characteristics; thus, such an approach is not convenient. There are exceptions, however, of dual system configurations with very small probe volumes, whose computed velocity-component variances are unbiased for at least two of the velocity components. One is a dual system with the lidars measuring at zero elevation angles; in this case the estimated $u$- and $v$-velocity variances are unbiased and the system is insensitive to the $w$-velocity variance. Another is a dual system with the lidars perfectly aligned with the wind direction and each lidar measuring at a given elevation angle; in this case the unbiased variances are those of the $u$- and $w$-components and the system is insensitive to the $v$-velocity variance.

As we normally use dual systems with low elevation angles, the alternative is to solve Equations (11) and (12) assuming, first, that $\hat{\varphi}(\mathbf{k} \cdot \mathbf{n}) = 1$ for both lidars, and, second, using the probe volume characteristics. Thus the velocity-variance bias can also account for the inherent bias that results from measuring with a dual-Doppler system.

## 3. Experimental Campaign

### 3.1. Description

The campaign took place at the Høvsøre Test Station in northwest Jutland, Denmark from 30 April to 15 May 2014 [29]. The goal of the campaign was to compare single- and dual-Doppler retrievals of the horizontal wind speed and direction. The single-Doppler retrievals were made by acquiring radial velocity measurements over a narrow sector with a long-range scanning lidar (WindScanner). For the dual-Doppler retrievals, two other WindScanners were configured to intersect their laser beams over a measurement volume of interest. The campaign is described in more detail in Simon [30]. The corresponding dataset can be accessed through Vasiljevic et al. [29]. A WindScanner is a modified version of Leosphere's WindCube 200S scanning pulsed Doppler lidar [31].

For this study, we are interested in the dual-Doppler measurements with the two WindScanners as they were deployed to stare at the 116.5-m cup anemometer and scanned with a pulse length of 200 ns accumulating Doppler spectra within 500 ms. Figure 4 illustrates the configuration of the two WindScanners and the meteorological mast at the site.

The two WindScanners $k$ and $w$ were measuring at elevation angles of 5.32 and 3.10° from a distance of 1135 and 1625 m, respectively, and at azimuthal positions so that winds coming from 165.66 and 229.57°, respectively, were aligned to their line-of-sights. Although they stared at the 116.5-m cup anemometer, we use the measurements from the 100-m Metek USA-1 sonic anemometer so that we can compare the radial velocity variances and the WindScanner reconstructed Cartesian-based velocity variances against those of the sonic anemometer. Hereafter, $u$ and $v$ refer to the wind-aligned and cross horizontal velocities, respectively.

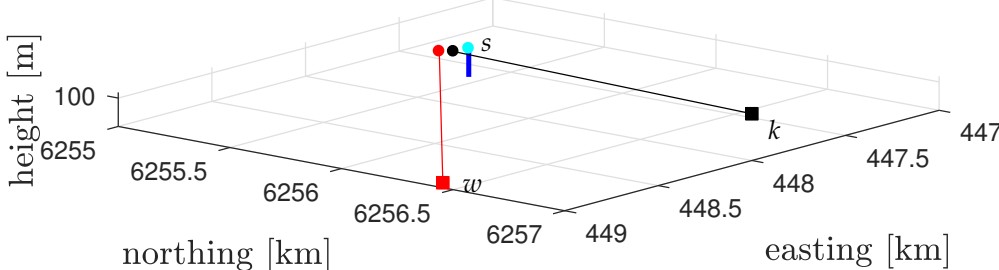

**Figure 4.** A sketch of the campaign at the Høvsøre Test Station in Denmark. The two WindScanners are shown in rectangles (*k* in black and *w* in red), their beams with similar color, the meteorological mast in blue with a 100-m sonic anemometer *s* in cyan. Coordinates are in UTM 32, WGS84.

### 3.2. Data Handling and Filtering

We performed the analysis of the measurements based on 10-min statistics. The two WindScanners were concurrently operating during 2582 10-min periods (approximately 15 days). The analysis was performed as follows:

1. We analyzed the 20-Hz sonic anemometer measurements within each 10-min period and filtered out the 10-min periods where more than 1% of the 20-Hz data within each 10-min showed a sonic status signal lower than two. This procedure resulted in 2574 10-min periods for the analysis.
2. We corrected the 20-Hz sonic anemometer measurements using the 3D correction by Metek GmbH [32], which has been shown to give the correct velocity-component spectra ratio in the inertial subrange [33].
3. We computed the mean horizontal wind speed and wind direction, linearly interpolated the voids in each 10-min time series due to the sonic status filter, performed azimuth and tilt rotations to the sonic anemometer velocity components to have the *u*-velocity component aligned with the 10-min mean wind direction, and calculated the *u*- and *v*-velocity variances.
4. We also transformed the sonic anemometer velocities to estimate the sonic-based radial velocities on the WindScanners beams' directions and computed the mean and variance of both radial velocities.
5. We retrieved the WindScanners' radial velocities within each 10-min interval and filtered out scans where the carrier-to-noise ratio (CNR) was either below $-25$ dB or above $-5$ dB. This procedure tries to avoid inaccurate scans and those where hard targets were hit (e.g., the mast). We only analyzed 10-min periods if both lidars showed at least 1000 scans per 10-min interval (out of $\approx$1200). This left us with 1982 10-min periods for analysis.
6. We used the algorithm by Goring and Nikora [34] to detect spikes in the radial velocity time series and filtered out those for each lidar. Again, we only analyzed 10-min periods if both lidars showed at least 1000 scans per 10-min interval (out of $\approx$1200). This left us with 1939 10-min periods for analysis.
7. We computed mean and variances on both WindScanners radial velocities, reconstructed the east and north velocity components from these radial velocities and thus the wind direction assuming $w = 0$. We rotated these velocity components to estimate *u* (aligned with the mean horizontal wind) and *v* the crosswind, and estimated their variances.

### 4. Results

Here we first provide theoretical estimations of the radial velocity spectra by the WindScanners and sonic anemometer for given turbulence characteristics and wind directions based on the experimental setup. Then we show comparisons between the measurements of the radial velocity variances and velocity-component variances between the WindScanners and the sonic anemometer measurements and their behavior with wind direction. We also add to the latter comparisons the estimations of the bias in radial velocity variances and velocity-component variances as a function

of wind direction and turbulence characteristics based on the Mann model. Finally, we show the theoretical spatial variation of the bias in velocity-component variances in a horizontal large domain.

### 4.1. Radial Velocity Spectra Estimation

We can estimate the theoretical behavior of the radial velocity spectrum of both the $k$ or the $w$ WindScanner beams and those of the sonic anemometer (aligned with either the $k$ or the $w$ WindScanner beam) for all possible wind directions using Equation (4). Figure 5 shows this behavior for the two wind directions parallel to either lidar, namely 165° for $k$ and 230° for $w$, assuming $z_R/L = 0.5$ for the WindScanners and $z_R/L = 0$ for the sonic anemometer, with Mann model parameters of $\alpha\epsilon^{2/3} = 0.1$ m$^{4/3}$ s$^{-2}$, $L = 100$ m, and $\Gamma = 3$ for both directions.

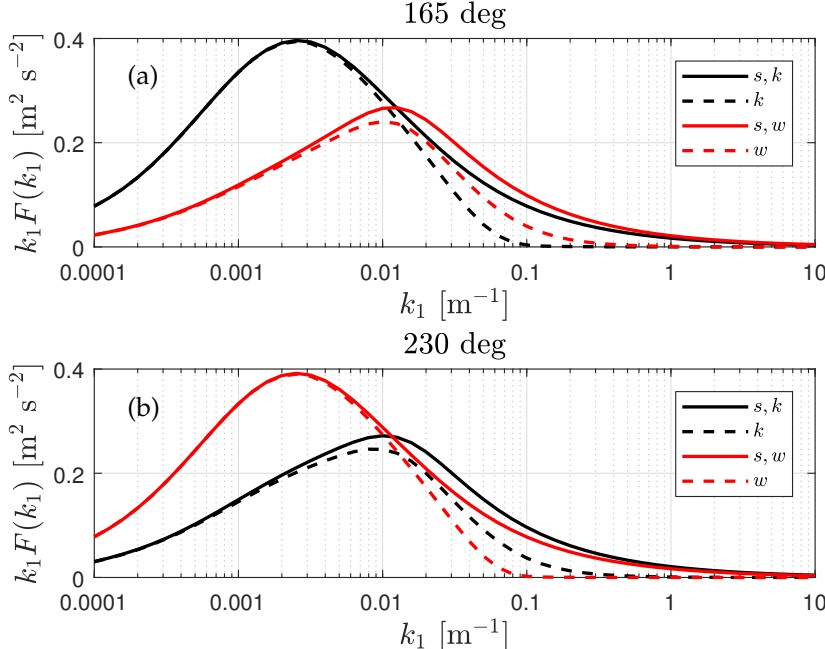

**Figure 5.** Radial velocity spectra of the $k$ and $w$ WindScanners and those of an ideal sonic anemometer aligned with either WindScanner ($s, w$ or $s, k$) for two wind directions: 165° (**a**) and 230° (**b**).

It is clearly illustrated that when the WindScanner beam is aligned with the wind direction, i.e., 165° for $k$ and 230° for $w$, the radial velocity spectrum peaks the highest, as it becomes close to the $u$-velocity spectrum since the elevation angle is rather small. The WindScanner beam that is not aligned with the direction shows a lower spectral peak, which is located at a higher wavenumber as this becomes more 'contaminated' by the $v$-velocity spectrum. It is also noticed the filtering effect, which attenuates the energy in the spectrum stronger with increasing wavenumbers and is strongest for the wind-aligned beam at the highest frequencies, but it is the other way around for the lowest frequencies.

### 4.2. Radial Velocities

We first start by looking at the ability of both WindScanners to perform radial velocity measurements by comparing their 10-min means with those from the sonic anemometers. Figure 6 shows such comparison and the plots illustrate two sets of data: the red indicating all radial velocity estimates and in black a selection of data where the difference between the WindScanner and the sonic anemometer measurement is lower than 0.5 m s$^{-1}$. Although we performed filtering on the radial velocities of both WindScanners, we still had 10-min periods where the difference in the mean between the WindScanners and the sonic anemometer was rather high, particularly for the $k$ WindScanner. This is mainly because the beam of $k$ hit often the mast, due to mast vibrations, and the CNR filter was not able to account for all hits. Since we are interested in turbulence measures, we use such a selected



data to avoid completely unrealistic turbulence measurements. The amount of 10-min periods left for analysis was therefore reduced to 1514.

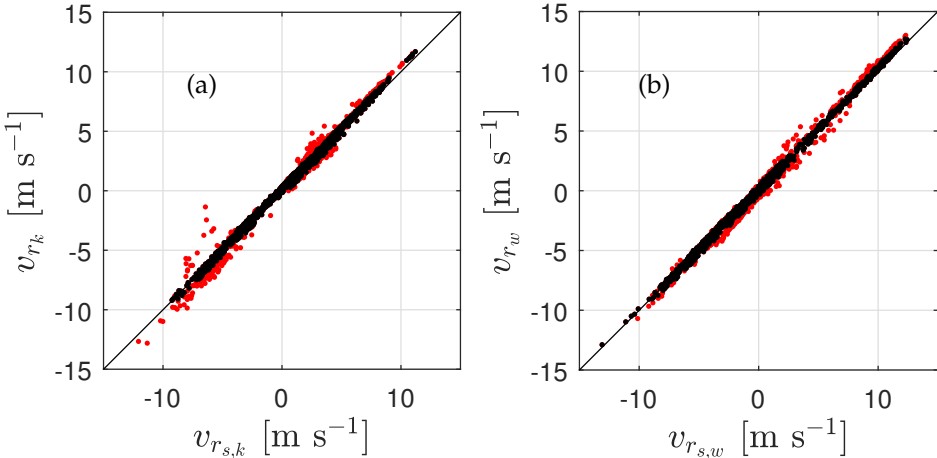

**Figure 6.** Scatter plot of the radial velocities of the WindScanners and the sonic anemometer in the beam direction of the $k$ lidar (**a**) and $w$ lidar (**b**). The red markers correspond to all data and the black markers to selected data (see text for details)

*4.3. Radial Velocity Variance*

When looking at the intercomparison between radial velocity variances between the WindScanners and the sonic anemometer (Figure 7), we find a high correlation and a bias of about 15% for both WindScanners. We know from Equation (4) that these biases are a function of the turbulence conditions and wind direction, and thus, high scatter does not necessarily indicate that the turbulence measurements are inaccurate but may indicate the variation in turbulence conditions inherent in the data.

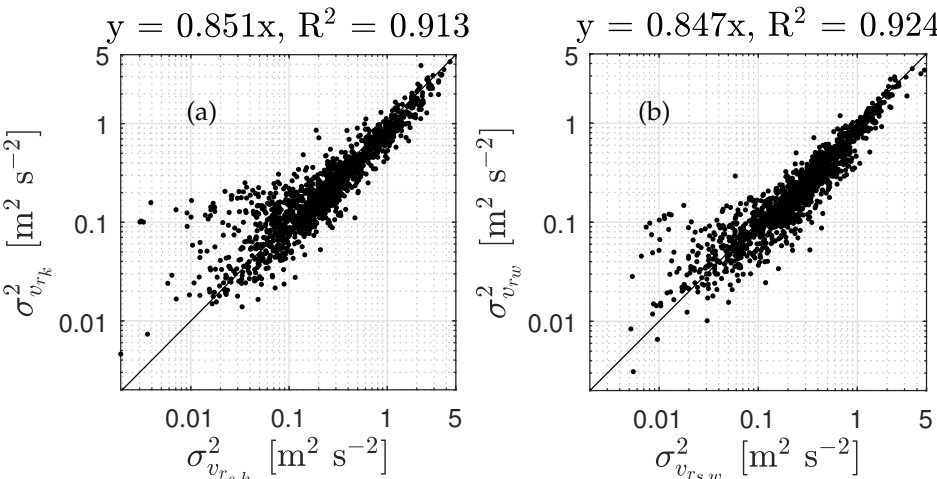

**Figure 7.** Scatter plot of the radial velocities' variances of the WindScanners and the sonic anemometer in the beam direction of the $k$ lidar (**a**) and $w$ lidar (**b**). The results of a linear regression through the origin are also shown together with the coefficient of determination $R^2$.

Figure 8 shows the behavior of such bias with wind direction, where we also plot a loess fit [35] with a 35% data span of the 10-min statistics and the theoretical predictions for $z_R/L = 0.25$, 0.5, and 1.0.

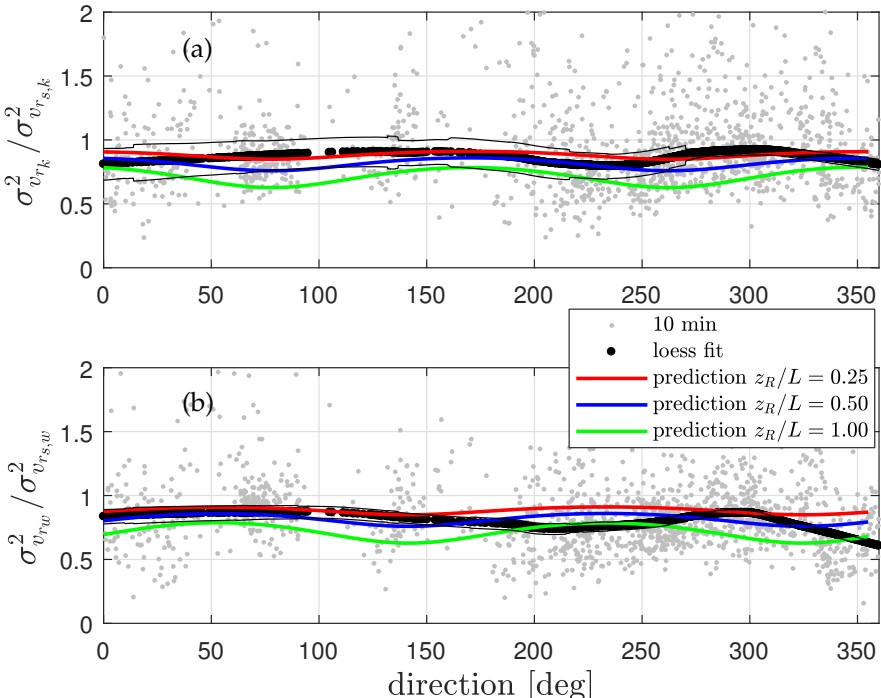

**Figure 8.** Ratio of the radial velocity variance of the WindScanners to the sonic anemometer as a function of wind direction in the beam direction of the $k$ lidar (**a**) and $w$ lidar (**b**). Measurements and a loess fit to the measurements are shown in markers and predictions in colored solid lines. The standard error of the loess fit is shown in solid black line.

For most wind directions, the bias in the measurements is below one but there are some 10-min periods in which it is above one for both WindScanners. This is theoretically not possible and thus the WindScanners' radial velocity variance might be overestimated (see Section 4.5 for a possible explanation). The bias is in average between the prediction assuming $z_R/L = 0.25$ and 0.5; some of the differences between predictions and observations are related to problems when estimating the variances from the measurements but they can also be due to the different turbulence characteristics of each 10-min period. For the 200–250° direction range, for both $w$ and $k$ lidars, the prediction using $z_R/L = 1.0$ fits the measurements indicating that under these directions the length scale of turbulence is comparable to the lidar probe volume.

### 4.4. Velocity-Component Variances

We can also look at the intercomparisons between velocity-component variances between the WindScanner system and the sonic anemometer. These are illustrated in Figure 9 where we get slightly higher correlations and lower biases (about 14%) when compared to the results for the radial velocity variance in Figure 7.

Since we know that these ratios of the velocity-component variances are also wind direction and turbulence dependent, we can also show their behavior with the measured direction (see Figure 10). The figure illustrates that these ratios are below and very close to one. The predictions for $z_R/L = 0.25$ and $z_R/L = 0.50$ match the measured average bias within some direction ranges closely well but, as with the radial velocity variance ratios, each 10-min period is characterized by different turbulence characteristics, and so we should not expect a perfect match for all measurements.

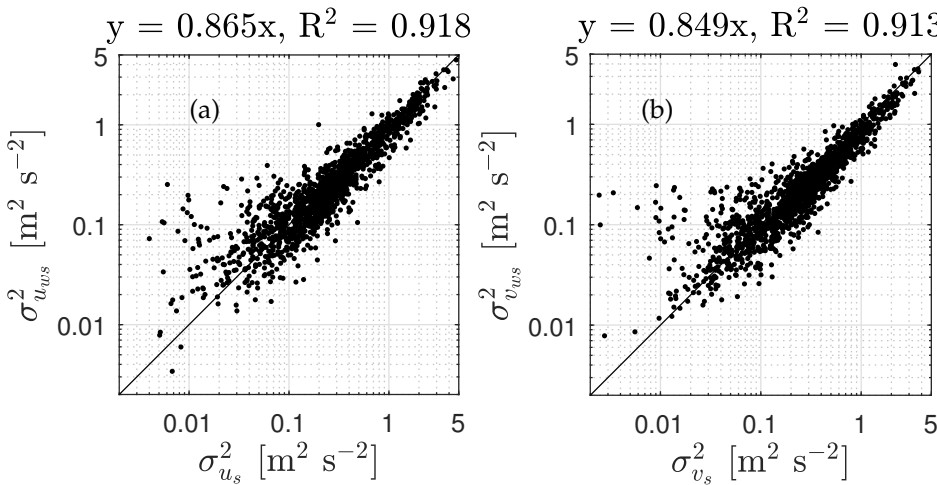

**Figure 9.** Scatter plot of the horizontal velocity-component variances of the WindScanners and the sonic anemometer. The results for the *u*- and *v*-velocity components are shown in frames (**a**) and (**b**), respectively. The results of a linear regression through the origin are also shown together with the coefficient of determination R$^2$.

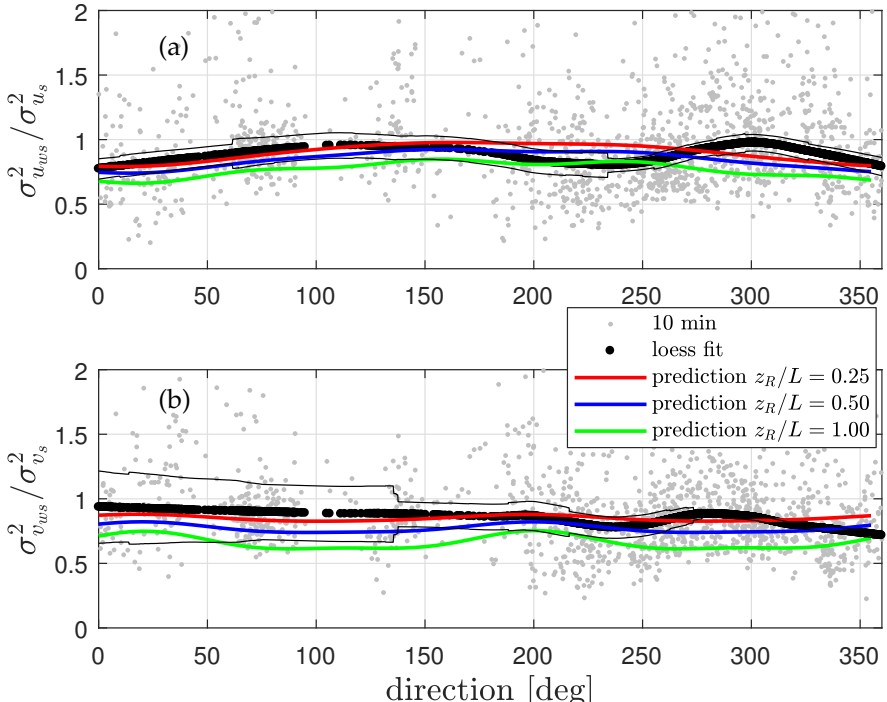

**Figure 10.** Ratio of the velocity components' variance of the WindScanners to the sonic anemometer as a function of wind direction. The result for the *u*- and *v*-velocity components are shown in frames (**a**) and (**b**), respectively. Measurements and a loess fit to the measurements are shown in markers and predictions in colored solid lines. The standard error of the loess fit is shown in solid black line.

### 4.5. Radial Velocity Spectra

In Figure 7, we show that particularly for very low values, the radial velocity variances of both lidars are often higher than those of the sonic anemometer. We know that this is not possible if turbulence is assumed homogeneous within the probe volume (see Figure 5); in this case the lidars only filter turbulence for a range of turbulence length scales. Figure 11 shows the ratio of the radial velocity spectra of the WindScanners' beams to the sonic anemometer for each of the WindScanners' beams. As illustrated, the spectra ratio decreases with increasing wavenumber, as expected due to

probe-volume filtering. However, at a frequency $\approx$0.2–0.3 Hz ($k_1 \approx$ 0.2–0.3 m$^{-1}$ for a mean wind speed close to $2\pi$ m s$^{-1}$), the ratio increases for both lidars, which is a typical behavior when lidar-based signals are contaminated by noise [28]. For completeness, we also show the theoretical spectra ratio for each WindScanner beam assuming $z_R/L = 0.25$ for a wind direction of 290°, which is close to the mean wind direction. Both observed ratios follow closely the theoretical computations up to $k_1 \approx 0.2$ m$^{-1}$. The theoretical ratios also show a difference between the two WindScanners, although lower than that of the observations.

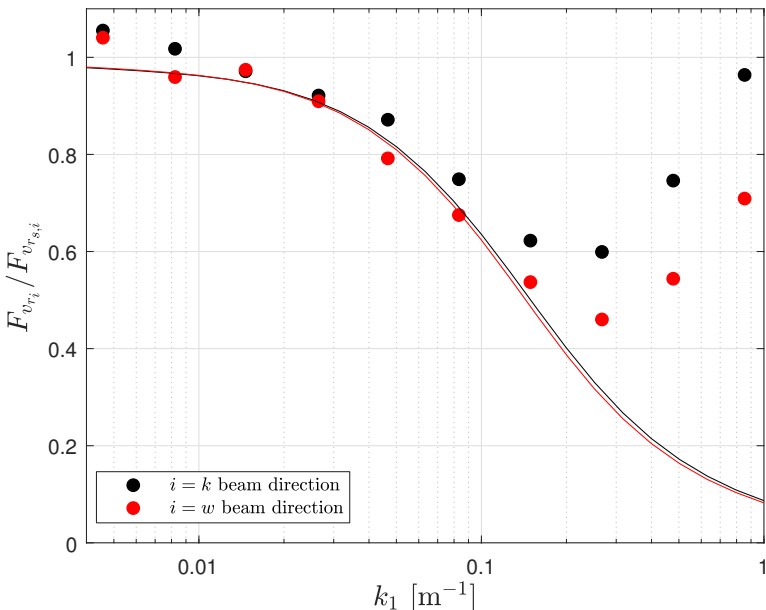

**Figure 11.** Radial velocity spectra ratio of the WindScanners to the sonic anemometer for each WindScanner beam as function of wavenumber. In solid lines, the theoretical spectral ratio is shown for each WindScanner beam for 290° winds assuming $z_R/L = 0.25$

### 4.6. Spatial Behavior of Velocity-Variance Biases

So far we know that the velocity variances computed from a dual-Doppler scanning system are biased when compared to those measured by an ideal instrument and that the bias depends on the turbulence characteristics and the scanning configuration with respect to the wind direction. This also means that if we are able to measure at multiple points within a spatial domain, i.e., a different beam configuration per position, we will then have a variation of the bias in velocity variances within the domain. This variation also depends on the turbulence characteristics and on the relative angles between the lidar beams and the wind direction.

Figures 12 and 13 illustrate examples for two wind directions of the variation of the velocity-variance biases within a horizontal domain using the locations of the lidars of the experimental campaign at Høvsøre. We assume that all points in the domain are measured at a height of 100 m and $z_R/L = 0.25$. We also constrain the analysis to all points within a rectangular domain where both lidars scan with elevation angles lower than 10°.

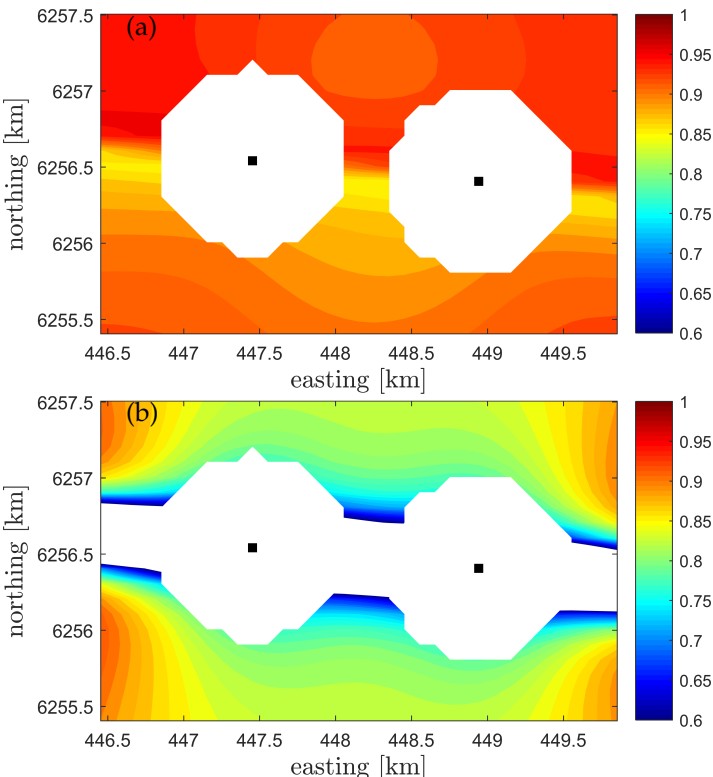

**Figure 12.** Spatial variation of velocity-variance biases of the dual-Doppler scanning system (lidars in solid rectangles) with respect to the velocity-variance measured at each point for a height of 100 m for a 90° wind. Turbulence is characterized by $\Gamma = 3$ and $z_R/L = 0.25$. Frame (**a**) shows the bias for the $u$-velocity component, i.e., $\sigma^2_{u_{ws}}/\sigma^2_{u_s}$, and (**b**) that for the $v$-velocity component, i.e., $\sigma^2_{v_{ws}}/\sigma^2_{v_s}$.

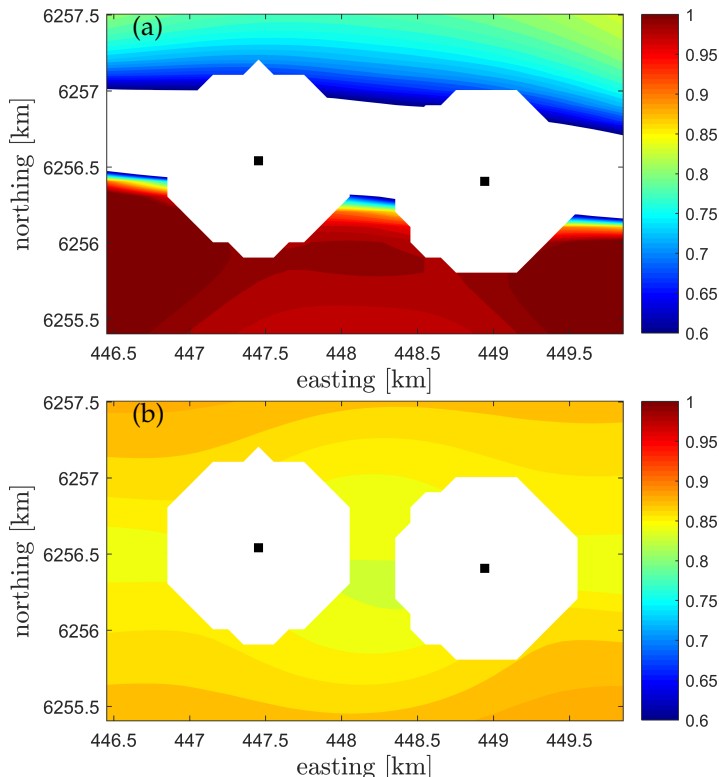

**Figure 13.** Same as Figure 12 but for a 180° wind.

In general, for the two wind directions, the biases in both velocity components are close to or below one, as expected from the results in Figure 10. For both directions, we see the difficulties to estimate both velocity variances when the beams are close to be aligned. Particularly for the 90° case and for the *v*-variance bias, we see a large area where the bias becomes lower than 0.6 (the range is limited to these values for visualization purposes and so it appears as white). This is because the lidars are deployed on a line close to 90° and it is difficult for the system to properly distinguish the variations of the *v*-component (nearly perpendicular to the beams). For the 180° case, we see that south and north of the lidars' connecting line, the bias is generally very close to one and much lower than one, respectively. This is because for this wind direction and for many of the positions, the lidar radial velocity variance becomes positively (south of the line) and negatively (north of line) contaminated by the *w*-component, in a similar fashion as shown in Figure 2. Note that the 10° constrain should appear as white disks around the lidar positions but in the figure the disks are ragged due to the spatial resolution (100 m) of the rectangular domain.

## 5. Discussion

It is unavoidable to ask ourselves the question on whether or not turbulence estimations are improved by adding a third or a fourth lidar. For the direct method, the system relating velocity variances and radial velocity variances becomes less singular the closer the added lidar(s) beam(s) are aligned with the vertical axis. For the indirect method, it would be very valuable as we would not be restricted to low elevation angles; a bias due to the probe volume still remains but as shown in the description of the method, it can be estimated for a multi-lidar system. Combining the abilities of both methods can be investigated in the future.

The direct method relies on our ability to determine the unfiltered radial velocity variance. For continuous-wave lidars, the Doppler radial velocity spectrum has been used to determine the unfiltered second-order moments of wind [26,28]. However, for pulsed lidars like most scanning lidars, the Doppler radial velocity spectrum has not been studied in much detail and few campaigns have attempted to store this information.

Spatial mapping of the velocity-variance biases, as that shown in Section 4.6, can help us for the planning of campaigns, particularly those related to estimation of site conditions and for the evaluation of turbulence models. Future work is undergoing on the analysis of the behavior of the velocity variances along the ridges at the Perdigão and Alaiz campaigns from the dual-Doppler scanning lidar measurements.

## 6. Conclusions

Although it was not the purpose of this study, from the experimental campaign at Høvsøre, we observed a high degree of agreement and no significant bias between the radial velocities measured by two WindScanners and those from a sonic anemometer deployed at 100-m on the site's meteorological mast.

We theoretically and experimentally demonstrated that the lidar radial velocity variance is filtered, i.e., it is lower than that of a point-like measurement, due to the lidar's probe volume. The degree of filtering is a function of the turbulence structure.

We also demonstrated both theoretically and experimentally that the variances of the velocity components from reconstructed velocity components can be filtered and (positively or negatively) contaminated by other velocity components.

We also showed that unfiltered velocity-component variances can be estimated from unfiltered radial velocity variances with a system of linear equations but that caution must be taken when using a dual-Doppler scanning system as the system can easily become singular.

Finally, we showed that the ratio of the true horizontal velocity variance to the reconstructed horizontal velocity variance measured by a number of WindScanners can be accurately estimated and

that this ratio depends on both the turbulence characteristics and the scanning configuration with respect to the mean wind direction.

**Author Contributions:** A.P. performed the analysis of the measurements, produced all results and figures, and drafted the manuscript. J.M. proposed the theoretical framework and revised the manuscript. All authors contributed to the methodology and finalization of the manuscript.

**Funding:** We would like to knowledge funding from Innovation Fund Denmark to the RECAST project (www.recastproject.dk).

**Acknowledgments:** We would like to acknowledge Nikola Vasiljevic for making the dataset publicly available and the Test and Measurements Section at DTU Wind Energy for helping in the execution of the experiment.

**Conflicts of Interest:** The authors declare no conflict of interest. The funders had no role in the design of the study; in the collection, analyses, or interpretation of data; in the writing of the manuscript, or in the decision to publish the results.

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
