# Peer review of "Turbulence Measurements with Dual-Doppler Scanning Lidars"

_remotesensing, doi:10.3390/rs11202444_

Round 1

Reviewer 1 Report

Review - 615998, Turbulence measurements with dual-Doppler scanning lidars.

The turbulent vortices inside cumulus clouds play an important role in formation, transport and dissipation of the energy inside powerful hurricane clouds.  The features of poly-phase turbulence   are of great interest to the scientific community now. There are many works for direct numerical simulations (DNS) of the wind velocity and temperature fields to examine the turbulent transport that presented at numbers of profile conferences. The motion inside vortex can be characterized by a superposition of numerous vortices and corresponding circular frequencies, which depend on wave-numbers k = 2pi/L related to ones from the inner or outer scales of clouds turbulence cell, in which L are 0.01 – 200 m typically (Monin 1971). The parameter of the turbulent energy dissipation is very important and interesting parameter for practice and it is related to energy transfer from the largest to minimal scales to become dissipated.

This paper uses and develops further the model of Mann to analyze the dual-Doppler lidar method. At the same time, authors present detailed description of the measurements using their algorithm and dual Doppler lidar system. By the way, the rays of both lasers can be easily increased in diameter for given point of space with the help of additional objective-lens that can be fixed near lidars. The comparison and coincidence of measurement's results through a sonic anemometers are demonstrated in series of figures. The variances of the wind velocity components can be detected and filtered to be presented for using. Considered paper by these authors is a next important stage in their work and in a series of their previous publications.

The authors developed an algorithm for using the system of two lidars for the experimental study of the components of the wind speed, and in particular for turbulence diagnostics. The proposed method of measurements can provide new information in the study of such important natural phenomenon as turbulence in atmospheric flow. I believe that the submitted article can be published in scientific journal.

Author Response

Dear Reviewer,

We are very thankful for your very positive review. Just for your knowledge, based on Review 3, we further emphasize on the novelty of our work both in the abstract and in the introduction of the paper.

Regards,
The authors

Reviewer 2 Report

This paper is very interesting for wind speed and direction measurement.

Author Response

Dear Reviewer,

We thank you for the very positive feedback. Just for your knowledge, based on Review 3, we further emphasize on the novelty of our work both in the abstract and in the introduction of the paper.

Regards,
The authors

Reviewer 3 Report

The authors address in detail the issue of direct and indirect computing of velocity-component variances  from lidar measurements using
information of the second-order statistics within the lidar probe volume.

The paper is technical addressing itself to remote sensing researchers using  dual-Doppler scanning lidars.

My concern is that the authors do not address the novelty of their contribution to state of the art in their domain i.e. what is original in their results,

This should be stated in both the abstract and in summary and conclusion section.

See for instance:

CHRIS G . COLLIER et al.:DUAL-DOPPLER LIDAR MEASUREMENTS FOR IMPROVING DISPERSION MODELS. BAMS 825-837.DOI: 10.1175/BAMS-86-6-825

Results from a field experiment involving the dual-Doppler measurements with the two WindScanners deployed to stare at the 116.5-m cup anemometer and scanned with a pulse length of
185 200 ns accumulating Doppler spectra within 500 ms are presented and analyzed.

The ability of both WindScanners to perform radial velocity
 measurements was assessed by comparing their 10-min means with those from the sonic anemometers.

Amongst many results they conclude with theoretically and experimentally demonstrating that the lidar radial velocity variance is filtered,
 i.e., it is lower than that of a point-like measurement, due to the lidar’s probe volume. The degree of
filtering is a function of the turbulence structure.

 In conclusion an interesting technical paper that needs minor revisions to highlight novel and original results and to separate those from results that were already known to remote sensing community,

I recommend acceptance of the ms subject to minor/medium revisions.

Author Response

Dear Reviewer,

We are very thankful for your very positive review. We believe that the novelty is inherent in the manuscript. However from your feedback to further emphasize it and to clarify the contents of the paper, we decided to add in the:

Abstract

A second line stating "Specifically, by using the Doppler radial velocity spectrum, one can estimate the unfiltered radial velocity variances."

The following lines were modified to highlight that this is novel approach "Here we show, for the first time, how to estimate such biases for a multi-lidar system and show, also for the first time, their dependence on the turbulence characteristics and the lidar beam scanning geometry relative to the wind direction."

The last sentence was also modified as follows "We also show that mapping these biases over a spatial domain aids at planning meteorological campaigns, where multi-lidar systems can potentially be used. Particularly, such maps aid for multi-point mapping of wind resources and conditions, which improve the tools needed for wind turbine siting."

Introduction

We added the following lines before the last paragraph:

"It is important to note that in most of the previous studies, the works concentrated in the analysis of the flow either by looking at the radial velocity (also known as the line-of-sight velocity), which is the basic output of a lidar scan, or at the velocity components, which were reconstructed from radial velocities using different methodologies. We believe that this is partly because understanding of atmospheric turbulence and how lidars probe the atmosphere are complex matters. We show, for the first time, how to compute the bias of velocity-component variances using radial velocity measurements from a multi-lidar system, in relation to those variances from an ideal anemometer. We also show for the first time the dependence of these biases on turbulence characteristics, scanning geometry, and lidar characteristics"

Regards,
The authors